# Chemical Composition and Flavor Characteristics of Cider Fermented with *Saccharomyces cerevisiae* and Non-*Saccharomyces cerevisiae*

**DOI:** 10.3390/foods12193565

**Published:** 2023-09-26

**Authors:** Yuzheng Wu, Zhigao Li, Sibo Zou, Liang Dong, Xinping Lin, Yingxi Chen, Sufang Zhang, Chaofan Ji, Huipeng Liang

**Affiliations:** SKL of Marine Food Processing & Safety Control, National Engineering Research Center of Seafood, Collaborative Innovation Center of Seafood Deep Processing, School of Food Science and Technology, Dalian Polytechnic University, Dalian 116034, China; 211720860000976@xy.dlpu.edu.cn (Y.W.); 211710832000932@xy.dlpu.edu.cn (Z.L.); 211710832000939@xy.dlpu.edu.cn (S.Z.); dongliang@dlpu.edu.cn (L.D.); yingchaer@163.com (X.L.); yingxichen24@163.com (Y.C.); zhangsf@dlpu.edu.cn (S.Z.); jichaofan@outlook.com (C.J.)

**Keywords:** cider, co-inoculated fermentation, *Saccharomyces cerevisiae*, non-*Saccharomyces* yeasts, GC-IMS, OVA

## Abstract

Cider flavor has a very important impact on the quality. Solid-phase microextraction-gas chromatography–mass spectrometry (SPME-GC-MS) combined with gas chromatography–ion mobility spectrometry (GC-IMS) tested different kinds of non-*Saccharomyces* yeasts and *Saccharomyces cerevisiae* (*S. cerevisiae*) co-inoculated for the fermentation of cider to determine differences in aroma material, and the determination of odor activity value (OAV) is applied less frequently in research. Through *Rhodotorula mucilaginosa*, *Debaryomyces hansenii*, *Zygosaccharomyces bailii*, and *Kluyveromyces Marxianus*, four different strains of non-*Saccharomyces* yeast fermented cider, and it was found that, in both the chemical composition and flavor of material things, compared with monoculture-fermented cider using *S. cerevisiae*, all differences were significant. Co-inoculated fermentation significantly improved the flavor and taste of cider. As in the volatile compounds of OVA > 1, octanoic acid (Sc 633.88 μg/L, co-inoculation fermented group 955.49 μg/L) provides vegetable cheese fragrance and decanoic acid, ethyl ester (Sc 683.19 μg/L, co-inoculation fermented group 694.98 μg/L) a creamy fruity fragrance, etc., and the average content increased after co-inoculated fermentation. Phenylethyl alcohol, which can produce a rose scent, was relatively abundant in cider samples and varied greatly among the groups. Moreover, the contents of ethyl lactate and 1-butanol in the Sc+Rm (ciders fermented by *S. cerevisiae* and *R. mucilaginosa*) were the highest of all of the cider samples. Different types of non-*Saccharomyces* yeast produced cider with different flavor characteristics. This study demonstrates that different species of non-*Saccharomyces* yeast do have an important impact on the characteristics of cider and that co-inoculation with non-*Saccharomyces* yeast and *S. cerevisiae* for cider fermentation may be a strategy to improve the flavor of cider.

## 1. Introduction

Cider is a beverage made from fermented apple juice that contains trace amounts of ethanol and is welcomed in many countries [1]. Cider flavor is an important index for evaluating cider quality [2]. Many factors influence the development of aromas in apple cider, for example, type of apple, maturity, fermentation condition, and yeast strain. Studies have found that yeast strains play an important role in the generation of flavor substances in cider [3].

In past studies, *Saccharomyces cerevisiae* (*S. cerevisiae*) has been used as the sole starter in cider brewing due to its ability to quickly convert sugar to alcohol. However, the resulting cider has a simple aroma and a sour taste [4]. This causes a reduction in people’s desire to purchase cider. Non-*Saccharomyces* yeast can synthesize a variety of enzymes that produce rich materials such as glycerin, turning raw material into esters and higher alcohols, with a variety of flavors and nutrients. It promotes the production of aroma components and is a good producer of fruit acetates [5]. However, it is intolerant to ethanol and often exists in the initial stage of fermentation, resulting in low fermentation efficiency [6]. A study found that the number of acetate acid, ethyl esters and acetate acid, isoamyl esters increased when *Williopsis saturnus* and *S. cerevisiae* were co-inoculated in wine [7]. Furthermore, another study found that the use of *Torulaspora delbrueckii* in co-inoculated fermentation with *S. cerevisiae* was an appropriate way to control flavor production during beer fermentation [8]. Therefore, it is of great importance to improve the quality of cider by co-culturing non-*Saccharomyces* yeast with *S. cerevisiae*.

The flavor of cider is an important factor representing its quality. The successful extraction of flavor substances is the key point of related substance research. At present, the main methods, such as headspace solid-phase microextraction [9], dynamic headspace gas chromatography, and solid-phase microextraction–gas chromatography–mass spectrometry (SPME-GC-MS) [10], are used for the analysis of flavor components in fruit wine. SPME-GC-MS has been widely used to analyze aroma compounds in many types of wine; it is an important research method for food flavor analysis at present. It can capture and detect high content of macromolecular flavor components with more than 10 carbon atoms. It can analyze specific flavor compounds of samples to be tested, with high detection sensitivity, convenient operation, and accurate results [11]. SPME-GC-MS combined with the odor activity value (OAV) can determine the ratio of the concentration of a single volatile on its odor threshold in water or air [12]. It is possible to study the potential key aroma compounds in cider and their contributions to the aroma of cider and to reveal the characteristic aroma patterns of cider. Meanwhile, gas chromatography-ion mobility spectrometry (GC-IMS) has been applied to discriminate subtle differences in the aroma of the product [13]. GC-IMS can capture and detect small molecular volatile components with 10 carbons or less so that substances with smaller molecular weights can be detected in the form of monomers and dimers, which is a modern visualization detection method that does not require pretreatment and is highly sensitive [14]. GC-IMS can be used as a supplement to SPME-GC-MS. To make the aroma analysis of volatile compounds in cider more comprehensive. At present, for SPME-GC-MS and GC-IMS combination, different non-*Saccharomyces* yeast and *S. cerevisiae* strains are tested in the fermentation of cider aroma components, and combining this with OVA difference analysis of cider aroma research is less common. 

Therefore, this study aimed to investigate the chemical composition and flavor characteristics of cider fermented by co-inoculation of several different non-*Saccharomyces* yeasts with *S. cerevisiae* and to obtain the important volatile compounds by SPME-GC-MS and GC-IMS, and OVA was used to analyze the aroma components of cider. The aim was to improve the quality of cider. This study utilized specific non-*Saccharomyces* yeasts (*Rhodotorula mucilaginosa* (*R. mucilaginosa*), *Debaryomyces hansenii* (*D. hansenii*), *Zygosaccharomyces bailii* (*Z. bailii*), and *Kluyveromyces marxianus* (*K. marxianus*)) and *S. cerevisiae* in combination, with the aim of providing basic support for its application in cider fermentation.

## 2. Materials and Methods

### 2.1. Strains and Fermentation Media

Five different yeast strains were applied, including one *Saccharomyces cerevisiae* and four non-*Saccharomyces* yeast species. *S. cerevisiae* No. 2.3854 (Sc) and *R. mucilaginosa* No. 2.2506 (Rm) were from the China General Microbiological Culture Collection Center (Beijing, China). *D. hansenii* No. 1714 (Dh) and *Z. bailii* No. 2.119 (Zb) were from the China Center of Industrial Culture Collection (Beijing, China). *K. marxianus* (Km) was from the Guangdong Microbial Culture Collection Center (Guangzhou, China).

The yeasts were inoculated into the sterile YPD medium (2% peptone, 2% dextrose, and 1% yeast extract) separately at 28 °C for 24 h, twice in succession, in the cider yeast inoculation quantity of 10^6^ CFU/mL. The yeast cells were obtained by centrifugating at 10,000 rpm for 10 min and suspending with sterile 0.85% (*w*/*v*) sodium chloride solution three times.

### 2.2. Cider Fermentation

Commercial apple juice was bought from a local supermarket in Dalian in Liaoning province (Huiyuan^®^ Apple 100% juice, China Huiyuan Juice Group, Beijing, China). The apple juice was clear apple juice. From the juice maker, the juice contained 10.2 g of total carbohydrates, 10 g of sugar, and 25 mg of sodium per 100 mL, without added sugar, preservatives, or flavoring. The sugar content of the juice was reached at 20° Brix by adding sucrose to the juice. Subsequently, a total of 400 mL of sterile apple juice (pasteurized at 104 °C for 15 min) was fermented in 500 mL sterile glass bottles. Fermentation was considered completed by monitoring the contents of residual sugars (<4 g/L) in cider. Apple juice that had not been inoculated with microorganisms was regarded as the blank sample (FAJ). Cider fermentation included a monoculture of *Saccharomyces cerevisiae* at a concentration of 10^6^ CFU/mL and co-inoculation of *Saccharomyces cerevisiae* and non-*Saccharomyces* yeast in a 1:1 ratio (1 represents 10^6^ CFU/mL). Fermentation trials were carried out at 28 °C in triplicate. The sugar content and alcohol content of the fermentation broth were measured once every 24 h and recorded. Until all data remained relatively stable, almost no gas was produced. The fermentation endpoint was reached. The samples at the end of the fermentation were centrifuged at 10,000 rpm for 10 min to obtain the supernatant and stored at −20 °C for later use.

### 2.3. Physicochemical Property Analysis

The pH was determined by a pH meter (METTLER TOLEDO FE28, Greifensee, Switzerland). Soluble solid content (SSC) was determined by a portable refractometer (HSU-32, Shanghai, China). Total acidity (TA) was titrated with 0.1 mol/L NaOH and expressed as the equivalent of tartaric acid. The ethanol concentration in the samples was determined according to the GB/T 15038-2006 method.

### 2.4. Glucose and Glycerol Content

The content of glucose was determined using a biochemical sensor measurement analyzer (SBA-90, Institute of Biology, Shandong Academy of Sciences, Ji’nan, China). The glycerol content was determined according to the instructions provided by the Glycerol Assay Kit (F005-1-1, Nanjing, China).

### 2.5. Organic Acid Content

Organic acids were determined by HPLC (Agilent 1220 Infinity II System, Agilent Technologies Inc., Palo Alto, CA, USA). The contents of tartaric acid, malic acid, acetic acid, citric acid, and succinic acid were determined using a ZORBAX-SB C18 column (5 μm, 4.6 × 250 mm) with a UV detector at 210 nm at room temperature. The injection quantity of the sample was 20 μL. The flow rate was 1.0 mL/min. Mobile phase A was diammonium hydrogen phosphate buffer (0.01 mol/L, pH = 2.8) and solvent B was a methanol solution. The elution gradient was 95:5 (*v*/*v*). The organic acid content was determined under these conditions, including malic acid, acetic acid, tartaric acid, succinic acid, and citric acid. Cider samples were injected after passing a 0.22 µm filter membrane. The standard curve was drawn on the basis of the concentration and peak area; each sample was set in parallel three times.

### 2.6. Volatile Analysis

#### 2.6.1. SPME-GC-MS Analytical Conditions

The volatile compounds were measured using SPME-GC-MS. A 2.0 mL sample of centrifuged cider was added to a 20 mL headspace injection vial (Supelco Inc., Bellefonte, PA, USA). At the same time, 20 μL of 4-methyl-1-amyl alcohol (120 mg/L) was added as an internal standard. Subsequently, the sealed injection vial was placed in a 60 °C water bath for 20 min stabilization and the solid-phase microextraction fiber (DVB/CAR/PDMS, Supelco, Inc., Bellefonte, PA, USA) was inserted into the vial and exposed to 60 °C headspace for 20 min. Extraction was performed after introducing Agilent 5977 a 7890 a fiber GC and MS (Agilent Technologies, Inc., Palo Alto, CA, USA) using a capillary column (HP-5MS, 30 m × 0.25 mm i.d, 0.25 μm film thickness). Conditions and methods were as referred to in previous studies [15].

#### 2.6.2. Parameters of GC-IMS Analysis

The volatiles were analyzed by gas chromatography–ion mass spectrometry equipped with a headspace sampling unit (FlavourSpec^®^, G.A.S., Dortmund, Germany). An MXT-WAX chromatographic column (30 m × 0.53 mm × 1 μm, Restek, Bellefonte, PA, USA) was used to separate volatile components. A 1 mL sample was added to a 20 mL headspace vial and incubated at 60 °C for 10 min before injection into the analysis. Nitrogen was used as the carrier gas. The column temperature was 60 °C and the IMS temperature was 45 °C. The incubation speed was 500 rpm. The flow rate of drift air was 150 mL/min, and the gas chromatographic conditions were as follows: 0 min, 2 mL/min; 2 min, 2 mL/min; 10 min, 10 mL/min; 20 min, 100 mL/min; and 30 min, 100 mL/min.

### 2.7. E-Tongue Analysis

The taste characteristics of the ciders were analyzed using a TS-5000Z type taste analysis system (Insent, Kanagawa, Japan). The electronic tongue sensor array is composed of five different taste sensors. The positive sensor consists of hybrid membranes, including the umami, saltiness, and sourness of three sensors, and the cathode sensor consists of a charged membrane, including the bitterness and astringency of two sensors. Bitter aftertaste, astringent aftertaste, and umami aftertaste (richness) can also be analyzed. The sensor and reference electrode were activated for 24 h prior to use. A 40 mL sample of tenfold diluted cider was placed into a measuring cup. The cider samples were then detected after the taste analysis system was self-tested successfully. Each sample was tested 4 times and the last 3 measurement data were adopted and analyzed.

### 2.8. Statistical Analysis

All experiments adopt a parallel experiment of three repeats. Data were analyzed by one-way analysis of variance (ANOVA) with SPSS 19.0 (SPSS Inc., Chicago, IL, USA), as well as Duncan’s post hoc test to test whether the differences between samples were statistically significant. Origin 2019 (OriginLab Corporation, Northampton, MA, USA) was used for the data plot. Redundancy analysis (RDA) was performed using Canoco for Windows v4.5 (Wageningen UR, Wageningen, The Netherlands). GC-IMS data were obtained and analyzed by VOCal software (v0.4.03) with three plug-ins from G.A.S., which include reporter software, gallery Plot software, and dynamic principal component analysis (PCA) software. The heat map was constructed using Heml vl.0 (Heatmap Illustrator, China, http://hemi.biocuckoo.org/down.php (accessed on 1 August 2023)).

## 3. Results and Discussion

### 3.1. Physicochemical Characteristics of Cider Fermented with Different Non-Saccharomyces Yeasts

The physicochemical characteristics, including pH, total acidity, alcohol content, glucose content, glycerol content, and total yeast count of the samples are shown in Table 1. Compared to the blank sample (FAJ), the apple juice inoculated with yeast had a higher pH value. There were significant differences in pH values between ciders inoculated with different non-*Saccharomyces* yeast (Table 1), which may lead to differences in some aromas [16]. The glycerin content also increased after inoculation compared to FAJ, which may be one reason why cider tastes smoother than apple juice. In terms of total acidity, Sc had the highest acidity, followed by the co-inoculation fermented group of *S. cerevisiae* and *R. mucilaginosa* (Sc+Rm). Among the various cider samples, Sc, the co-inoculation fermented group of *S. cerevisiae* and *Z. bailii* (Sc+Zb) and Sc+Rm had the highest total yeast count, and there were no significant differences between them. These were followed by the co-inoculation fermented group of *S. cerevisiae* and *D. hansenii* (Sc+Dh) and by the co-inoculation fermented group of *S. cerevisiae* and *K. marxianus* (Sc+Km); this may be caused by contact between yeast cells, which, in turn, produces an inhibitory effect [17]. Generally, a lower alcohol concentration was found in the co-inoculation fermented groups, except Sc+Rm. As shown in Table 1, all the samples inoculated with yeast had a low glucose concentration, indicating that the yeasts consumed a large amount of glucose (all consumption rates were greater than 98%). Furthermore, the glucose consumption of the co-inoculation fermented groups was higher than that of the Sc group.

The results of organic acids in cider are shown in Figure 1. Five organic acids were detected in the cider, of which acetic acid was the main organic acid in the cider (Appendix A). The acetic acid content in the co-inoculation fermented groups was higher than Sc and FAJ; a similar phenomenon was also observed in previous studies [1]. The highest content of malic acid was found in the FAJ group (1.14 g/L), and the content of malic acid in the Sc group was significantly higher than that in the co-inoculation fermented groups but lower than in the FAJ group. The reason for this result may be that malic acid can be used as a carbon source for yeast, which is consumed and utilized [18]. Furthermore, the succinic acid content in the Sc group was higher than in the co-inoculation fermented groups, which may cause a bitter taste [19].

### 3.2. Volatile Compounds of Cider Fermented with Different Non-Saccharomyces Yeasts

#### 3.2.1. SPME-GC-MS Analysis of Cider Fermented with Different Non-*Saccharomyces* Yeasts

Appendix A shows the concentrations of volatile compounds in cider detected by GC-MS. A total of 58 kinds of volatile compounds were detected in the cider samples, including 24 esters, 6 alcohols, 12 acids, 4 aldehydes, 1 ketone, and 11 other compounds. The highest volatile content among them was most types of esters, alcohols, and acids. Our study found several previous important aroma esters identified in ciders, such as hexanoic acid, ethyl ester, octanoic acid, ethyl ester, decanoic acid, ethyl ester, and acetic acid, 2-phenyl ethyl ester [20,21]. In most of the co-inoculation fermented ciders, several of the esters (except hexanoic acid, ethyl ester) were higher than Sc, which proved that inoculation with *R. mucilaginosa*, *D. hansenii*, or *Z. bailii* could improve the contents of these compounds. Interestingly, (Z)-3-Hexen-1-ol was detected only in FAJ. 3-methy-1-butanol was relatively abundant in several cider samples, except Sc+Rm and Sc+Km; the difference in content between the groups was great. The material can produce a smell similar to that of nail polish [22]. 2-Furanmethanol (3.91 ± 5.53 μg/L) was detected only in Sc+Rm, which has the aroma of coffee and caramel. 1-Dodecanol (3.00 ± 4.25 μg/L) was detected only in Sc+Km, which has the aroma of violet. Regarding phenylethyl alcohol, an important aroma component identified in apple wine [23], it can give the cider a rosy aroma [24]; a high concentration was found in cider from co-inoculation fermented groups (except Sc+Km), followed by that from monoculture fermentation. The range of concentration was 488.65~782.68 μg/L and there was a significant difference among the groups. The result was slightly different from previous research, which may be caused by using different species of non-*Saccharomyces* yeast [5]. Hexanal and furfural were detected only in FAJ, 2,4-dimethyl-benzaldehyde only in Sc+Zb and Sc+Km, and 2,5-dimethyl-benzaldehyde only in Sc and Sc+Zb; no aldehydes were detected in Sc+Dh.

The total concentration of volatile flavor compounds in cider samples is shown in Figure 2A. The total content of volatile compounds in cider was significantly higher than that in FAJ, and the total content of volatile compounds in co-inoculation fermented cider (except the Sc+Km group) was higher than that in Sc. To clearly observe the differences in volatile compounds between the cider samples, the data obtained by GC-MS were used to prepare a heatmap, as seen in Figure 2B; the color (from blue to red) indicates that the relative content changed from low to high. The volatile flavor components of several cider samples were significantly different in species and contents, which may be caused by different species of non-*Saccharomyces* yeast [5]. There were more volatile compounds in the yeast-inoculated samples than in FAJ, indicating that microorganisms produced many metabolites during fermentation. Furthermore, ciders produced by co-culture fermentation had more species of volatile compounds than Sc; this result was in accordance with a previous study [25]. As shown in Figure 2C, the total concentrations of esters, alcohols, and acids in different cider samples were different. The total concentration of esters in FAJ without yeast inoculation was significantly lower than that of other cider samples. Except Sc+Km having the highest total ester concentration, there was no significant difference between the total ester concentration in the co-inoculation fermented groups and the Sc.

Based on relevant reports on the aroma threshold [26,27], the aroma values of the odor activity values (OVA) were calculated and the characteristic aroma components were determined. OVA and aroma description reference the predecessors’ research [23,26,28]. OVA is the ratio of the amount of a compound to the aroma threshold of that compound. An OVA greater than 1 is the aroma component that is of great importance in the flavor of apple juice or cider, which is called the characteristic aroma component of apple juice or cider. The contribution to the aroma of cider will increase with increasing OVA.

As shown in Appendix A, there were certain differences in OVA between apple juice and the five fermentation groups. FAJ has a great difference compared to other fermentation groups. There were 16 aroma substances with an aroma value greater than 1. 1-Butanol, 3-methyl-, 3-hexen-1-ol, (Z)-, hexanal, furfural, butanoic acid, 2-methyl-, ethyl ester, butanoic acid, 3-methyl-, ethyl ester, decanoic acid, ethyl ester, hexanoic acid, octanoic acid, and dodecanoic acid had high aroma values and significantly affected the overall aroma of the six groups of samples. The 3-hexen-1-ol, (Z)-, hexanal, furfural and butanoic acid, and 3-methyl-, ethyl ester aroma component imparted grassy, spicy, and almond flavors to the FAJ group only. After fermentation, the flavorful substance was not detected (3-hexen-1-ol, (Z)-, hexanal, furfural and butanoic acid, and 3-methyl-, ethyl ester). Hexanoic acid, ethyl ester and acetic acid, 2-phenyl ethyl ester were detected in other fermentation groups, except for FAJ; they give the cider its green apple flavor and sweet honey flavor. The aroma value of the co-inoculation fermented groups was higher than that of the Sc group. It can be found through the research results that the fermentation of apple juice could improve the bad flavor components, and co-inoculation fermentation could improve the flavor of apple wine. 1-Butanol, 3-methyl- can endow the sample with a mild flavor. After fermentation, the aroma value of 1-butanol, 3-methyl- in the Sc group is significantly increased. The Sc+Zb group acetic acid, 2-phenylethyl ester and octanoic acid, ethyl ester compound with a sweet smell has a higher aroma value. The aroma value of 1-butanol, 3-methyl- in the co-inoculation fermented groups is significantly lower than that in the Sc+Zb group, except for the Sc+ZB group, and the aroma value of this substance is the lowest in the Sc+Km group. Sc+Km was found to contain higher alcohols, and 1-dodecanol (OVA > 1) with a sweet mellow fragrance was detected only in Sc+Km. Octanoic acid, ethyl ester had apple notes characteristic of cider, decanoic acid, ethyl ester had rum-like creamy and fruity notes, and octanoic acid, ethyl ester and decanoic acid, ethyl ester had higher OVA values in Sc+Dh than in the other fermentation groups. Acetic acid, 2-phenylethyl ester with a sweet honey aroma in the co-inoculation fermented groups had the lowest OVA value in the Sc+Rm group. The contents of octanoic acid, n-decanoic acid, and dodecanoic acid were higher in the cider inoculated with yeast. They had a vegetable cheese flavor; a pleasant rancid, sour, fatty, citrus flavor; and a mild, fatty, coconut, bay oil flavor. The contents of octanoic acid, n-decanoic acid, and dodecanoic acid were higher in the Sc+Dh group than in the other groups.

#### 3.2.2. GC-IMS Analysis of Cider Fermented with Different Non-*Saccharomyces* Yeasts

As shown in Appendix A, 29 volatile compounds were identified in cider samples by GC-IMS, including 14 esters, 8 alcohols, 3 aldehydes, 3 ketones, and 1 acid, which proved esters and alcohols were the main contributors to the cider aroma. If the concentration of volatile compounds is relatively high, monomers (M) and dimers (D) can be produced [29]. 

Cider samples from citron samples from the topographic map are shown in Figure 3A,B, in which the migration time of the ions is plotted in the abscissa and the retention time of gas chromatography is plotted in the co-ordinate. The background of the topographic plots is blue, and the reactive ion peak abscissa (RIP) appears at 1.0. The points on either side of the RIP represent volatile compounds; the redder the dot, the higher the concentration. As shown in Figure 3A, there were large differences between FAJ and other yeast-inoculated ciders, which may indicate that yeasts are important for cider fermentation. However, the volatile compounds in the ciders were similar, except for FAJ. Sc topographic map was used as a reference, and topographic maps of other samples were subtracted to obtain the difference comparison model. If the concentrations of the compounds are similar, the background is white. The closer the color is to red, the higher the concentration is compared to the reference concentration and, for blue, vice versa. The results are shown in Figure 3B. FAJ is different from the five ciders, but the ciders were similar among them. The Sc+Zb group had the lowest concentration of aroma substance among the co-inoculation fermented groups. The aroma substance concentration of the Sc+Dh group was significantly different from that of the other co-inoculation fermented groups. 

Comparison of the fingerprints of the obtained peaks allowed the identification of differences in specific volatile compounds. The results are shown in Figure 3C. The concentration of the substances in the yellow frame had apparent differences between FAJ and other ciders; they were much higher in other ciders than those in FAJ and they were also different from each other. Regarding ethyl lactate and 1-butanol, the contents of these two compounds in Sc+Rm were slightly higher than the other four groups of cider. 1-Butanol has a typical nasal aroma, while ethyl lactate has a faint fruity, fatty aroma. For acetic acid, 2-phenyl ethyl ester and octanoic acid, ethyl ester, the contents of those two compounds in Sc were the lowest of the five groups of ciders (except for the FAJ). The contents of 2-methyl-1-propanol, 3-methyl,1-butanol, acetone, and ethanol had little difference in the five cider groups; the content of the Sc group was the lowest. There were some volatile compounds in FAJ that had higher contents than those of other cider samples, such as hexanal, cis-3-hexenyl acetate, furfural, and 3-pentanone. The black frame content was lower in Sc and FAJ cider than in other samples. The substances in the black box include ethyl butyrate, which can provide the fruit aroma of cider [23], indicating that the fruit aroma of the co-culture fermentation cider is more intense. The concentration of red frame substance was higher in FAJ but lower in other ciders. Except for 1-hexanol, the concentration of red frame substance differed slightly in the fermentation group.

Figure 3D is the cluster analysis plot of the cider samples. According to the plot, the FAJ group clustered into one category, the Sc group and the Sc+Zb group clustered into one category, and the rest of the co-inoculation fermented groups (Sc+Dh, Sc+Km, and Sc+Rm) clustered into one category.

#### 3.2.3. Comparison of the Results of SPME-GC-MS and GC-IMS

According to Appendix A, the concentrations of alcohols, esters, and acids in apple cider are high, except in FAJ, such as phenylethyl alcohol, acetic acid, hexyl ester, and acetic acid, indicating that these substances are the basic components of apple cider aroma. There were some volatile compounds that were detected only by GC-IMS, such as ethyl lactate, isoamyl acetate, ethyl butanoate, 2-methylpropyl acetate, 2-pentanone, ethyl propanoate, hexanoic acid, ethyl ester, and 1-hexanol. This was due to the high sensitivity of GC-IMS [30]. Furthermore, some volatile compounds could not be identified due to the incomplete database. Therefore, the application of a combination of GC-MS and GC-IMS to detect volatile compounds in cider samples could be more comprehensive and accurate.

#### 3.2.4. Combination of SPME-GC-MS and GC-IMS Results for Multiple Analysis of Volatile Compounds

PCA was used to show the distribution of the cider samples. In Figure 4A, the principal components (PCs) PC1 and PC2 represent 95.3% and 1.8% of the total variance, respectively, so the two PCs elucidated 97.1% of the total variables in this model. This result showed that they were able to explain the total variance analysis. As shown in Figure 4A, three groups were observed, which indicated that there were significant differences in flavor between the FAJ, Sc, and co-inoculation fermented groups. As shown in Figure 4B, volatile compounds with VIP > 1 were isoamyl acetate, ethyl hexanoate, 3-methyl-1-butanol, acetic acid, ethyl 3-methyl butanoate, ethanol, ethyl 2-methyl butanoate, octanoic acid, ethyl ester, 1-butanol, 3-methyl-, 1-hexanol, ethyl butanoate, 2-methyl-1-propanol, acetone, and butan-1-ol. This indicates that they have an important effect on the flavor of cider. This is consistent with the result of previous research [2].

RDA was used to analyze the correlation between the physicochemical properties of cider and volatile compounds. As shown in Figure 4C, FAJ was positively correlated with malic acid, tartaric acid, citric acid, and glucose. The Sc group was positively correlated with succinic acid, total acid, glycerol, and alcohol content. Among them, Sc showed a strong correlation with total acidity and succinic acid. The highest total acidity and succinate were found in Sc (Table 1, Figure 1). The succinic acid taste is strong, bitter, and salty, which can make wine taste strong and increase the mellow feeling. The positive correlation of Sc with glycerol and alcohol was weak. The positive correlation of Sc with glycerol and alcohol was weak. Some studies have found that, during the fermentation process of cider, a part of the alcohol can be converted into acids. While yeast uses sugar to produce alcohol, it also produces other by-products, such as glycerol, higher alcohols, and organic acids [31]. These make cider sweet with acid, and acid is also astringent. Meanwhile, Sc+Zb and Sc+Rm of the co-inoculation fermented groups were also affected. In the process of co-culture fermentation, the growth of non-*Saccharomyces* yeast is inhibited by *S. cerevisiae* and this further influenced the non-*Saccharomyces* yeast in the cider fermentation process in terms of growth and metabolic activity [17]. The co-inoculation fermented groups were positively correlated with acetic acid content. Acetic acid is the main volatile acid in cider; it can enrich wine’s taste and aroma. A diagram showed the positive correlation between acetic acid and pH. Fruit wine contains higher alcohols and aldehydes, both of which are oxidized so that a large amount of acetic acid is accumulated in fruit wine. Acetic acid will further esterify with alcohols, which will significantly increase the concentration of fruity esters, such as acetic acid, isoamyl ester and acetic acid, 2-phenyl ethyl ester, and endow fruit wine with special fragrance [32]. In the wine fermentation process, organic acid and alcohol esterification reaction, besides giving wine an ester aroma fragrance compound, can also lead to a higher alcohol content drop [33]. This is consistent with the results of the present study. According to Figure 4C, the co-inoculation fermented groups were positively correlated with the content of most volatile compounds, the Sc group was positively correlated with some volatile compounds, and FAJ was only correlated with a small number of volatile compounds.

Each yeast has its characteristics, and the differences in the activities of each enzyme in the glycolysis process of different yeasts make each strain different, leading to differences in the metabolism of sugar, acid, glycerol, and aroma components [34,35]. The characteristics of the different strains and the interactions between the strains, as well as the combined effects between substances, give cider its complex flavor characteristics.

### 3.3. E-Tongue Analysis of Cider Fermented with Different Non-Saccharomyces Yeasts

Data collected with the electronic tongue of five cider samples were analyzed using a radar fingerprint (Figure 5). Significant differences can be observed between cider samples in sourness, astringency, umami, and saltiness. Sc had the highest sourness and saltiness value, while Sc+Dh had the lowest sourness value. Moreover, Sc+Dh had the highest umami value. In terms of astringency, Sc+Km had the lowest astringency value, while Sc+Zb had the highest.

## 4. Discussion

Wine yeast strains were added through co-inoculation fermentation to several cider samples to study their influence, having the following findings. Modulatory effects on physicochemical properties vary depending on the microorganisms utilized. The pH and glycerol of apple cider after inoculation were higher than those of apple juice. Sc has the highest acidity. In addition to Sc+Rm, the ethanol concentration of the co-inoculation fermented groups is low. Acetic acid is the major organic acid in cider. The acetic acid content of the Sc and FAJ groups was significantly lower than that of the co-inoculation fermented groups. Malic acid content was reduced in the co-inoculation fermented groups compared to that in the Sc group.

In the aroma spectrum of cider samples, esters, alcohols, and acids are the most abundant and diverse volatile compounds in the samples. The contents of the corresponding substances of the main volatile esters, alcohols, and acids in cider are absent or very low in FAJ, such as decanoic acid, ethyl ester, phenylethyl alcohol, and octanoic acid. Phenylethyl alcohol, which can produce a rose scent, was relatively abundant in cider samples and varied greatly among the groups. This indicates that the aroma of cider is mainly produced in the fermentation process of yeast and has little relationship with the volatile aroma components of the raw material itself. This is consistent with the results of previous studies [36]. According to Schreier P, apples contain non-volatile glycoside precursors of substances such as phenylethyl alcohol. When apples are broken, the glycosides of phenylethyl alcohol can be hydrolyzed by plant enzymes or by enzymes produced by yeast during fermentation. These release the aroma of phenylethyl alcohol. Alternatively, phenylethyl alcohol may also be formed by yeast during fermentation, capable of synthesizing phenylethyl alcohol from substrates present in apples. The contents of ethyl lactate and 1-butanol in the Sc+Rm were the highest of all of the cider samples. Total acids and esters were highest in Sc+Dh, and total acid concentrations were significantly higher in Sc+Dh and Sc+Rm than in Sc. This indicates that the yeast species have an important effect on cider fermentation.

There were 16 aroma substances with an aroma value greater than 1.0. The contents of hexanoic acid, ethyl ester and octanoic acid, ethyl ester in the Sc were the lowest among the five groups of ciders. The OVA of octanoic acid, ethyl ester with apple fragrance and decanoic acid, ethyl ester with cream fragrance in the Sc+Dh group was higher than in the other fermentation groups. Octanoic acid and acetic acid have a vegetable cheesy flavor and lauric acid has a mild fatty, coconut, bay oil flavor. The content of octanoic acid, acetic acid, and dodecanoic acid (OVA > 1) in the Sc+Dh group was higher than in the other cider samples. We found that the aroma of the cider fermented by the *D. hansenii* strain is characterized by a mild creamy aroma. The aroma values of the co-inoculation fermented groups were mostly higher than those of the Sc group. The difference between aroma types is that the cider fermented by *Z. bailii* has a prominent sweet aroma, the apple wine fermented by *R. mucilaginosa* has a strong hops aroma, the apple wine fermented by *K. marxianus* has a strong mellow and sweet aroma, and the apple wine fermented by *D. hansenii* has a rich milk aroma. These differences may be due to the different types of yeast in the fermentation of different levels of raw materials for apples in the process of adsorption, oxidation, hydrolysis, and polymerization [32]. The contents of most volatile flavor compounds were positively correlated with the co-inoculation fermented groups. Each yeast, according to its characteristics, leads to differences in metabolic substances and aroma components. Studies have found that the fermentation of apple juice can improve the undesirable flavor components, and co-culture fermentation can enhance the aroma of apple wine. 

There are significant differences in the taste of different samples of cider, and co-culture fermentation can improve the acidity and richness of cider. 

## 5. Conclusions

The effects of several non-*Saccharomyces* yeast strains on co-culture fermentation of cider samples were studied. The adjustment role of physical and chemical properties for use varies among different microbes. Glucose was made full use of, while alcohol was between 9 and 11% (*v*/*v*). Co-culture fermentation significantly reduced the malic acid content in the cider. In the aroma spectrum of cider samples, esters, alcohols, and acids contributed significantly to the aroma of cider. The wine yeast strains have important effects on apple wine aroma and the different kinds of *Saccharomyces* yeast used in co-culture fermentation of cider produced aroma differences. There were also significant differences in taste between cider samples. Fermentation of apple juice can improve the undesirable flavor components, and co-culture fermentation can improve the flavor and taste of apple wine.

This study indicates that different species of non-*Saccharomyces* yeast do have an important influence on the characteristics of cider. This conclusion may provide a little insight into exploring the use of different species of non-*Saccharomyces* yeast and the fundamental basis for the use of co-inoculation fermentation in cider fermentation.

## Figures and Tables

**Figure 1 foods-12-03565-f001:**
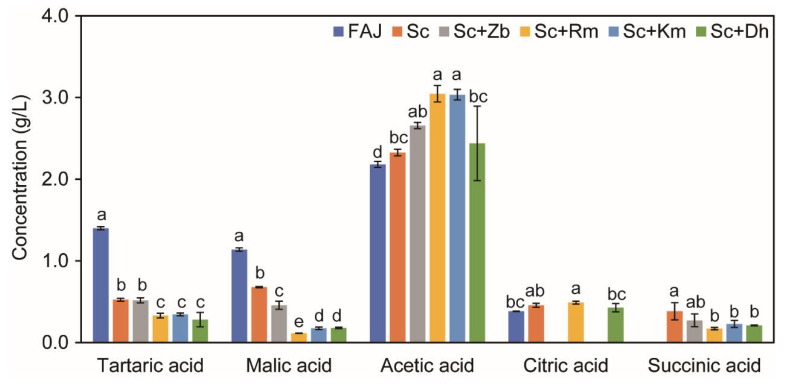
The concentration of organic acid in apple juice and cider samples fermented with different non-*Saccharomyces* yeast. Results represent the mean ± SD for three independent experiments. Different letters represent significant differences at the 95% confidence level (Duncan’s test). Sc, *S. cerevisiae* monoculture. Sc+Zb, *S. cerevisiae* and *Z. bailii* co-inoculation. Sc+Rm, *S. cerevisiae* and *R. mucilaginosa* co-inoculation. Sc+Km, *S. cerevisiae* and *K. marxianus* co-inoculation. Sc+Dh, *S. cerevisiae* and *D. hansenii* co-inoculation. FAJ, apple juice.

**Figure 2 foods-12-03565-f002:**
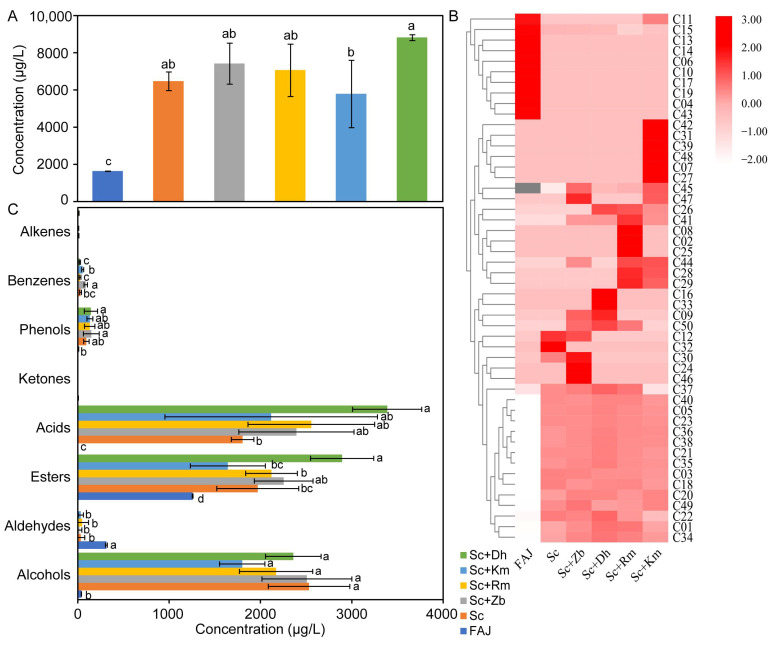
The total concentration (**A**), heatmap (**B**), and classification (**C**) of volatile flavor compounds identified by GC/MS in apple juice and cider samples fermented with different non-*Saccharomyces* yeasts. Results represent the mean ± SD for three independent experiments. Different letters represent significant differences at the 95% confidence level (Duncan’s test). Sc, *S. cerevisiae* monoculture. Sc+Zb, *S. cerevisiae* and *Z. bailii* co-inoculation. Sc+Rm, *S. cerevisiae* and *R. mucilaginosa* co-inoculation. Sc+Km, *S. cerevisiae* and *K. marxianus* co-inoculation. Sc+Dh, *S. cerevisiae* and *D. hansenii* co-inoculation. FAJ, apple juice.

**Figure 3 foods-12-03565-f003:**
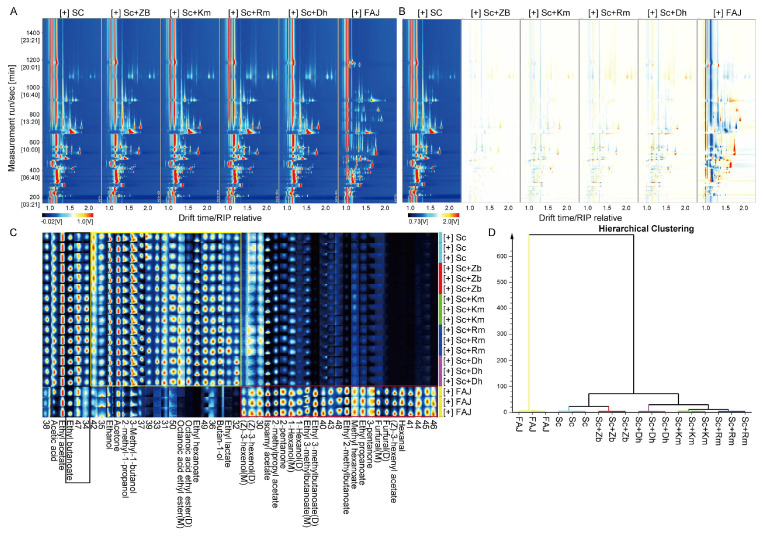
The topographic plot (**A**), comparison results under the spectral diagram (**B**), gallery plot (**C**), and clustering analysis (**D**) of volatile flavor compounds identified by GC/IMS in apple juice and cider samples fermented with different non-*Saccharomyces* yeasts. (**B**) Sc sample was selected as the reference. A zone of each topographic plot, which contains most of the important data, is labeled with a rectangle (black, yellow, and red), respectively. Sc, *S. cerevisiae* monoculture. Sc+Zb, *S. cerevisiae* and *Z. bailii* co-inoculation. Sc+Rm, *S. cerevisiae* and *R. mucilaginosa* co-inoculation. Sc+Km, *S. cerevisiae* and *K. marxianus* co-inoculation. Sc+Dh, *S. cerevisiae* and *D. hansenii* co-inoculation. FAJ, apple juice.

**Figure 4 foods-12-03565-f004:**
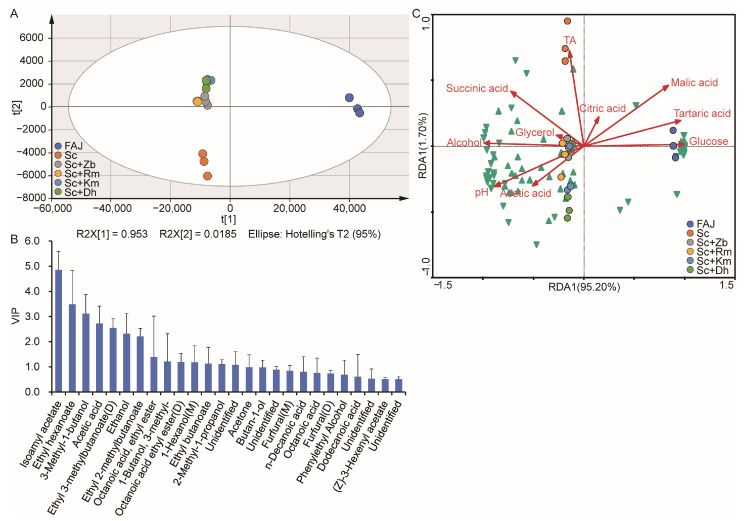
The PCA analysis (**A**), VIP plot (**B**), and RDA analysis (**C**) of volatile flavor compounds identified by GCMS and GC/IMS in apple juice and cider samples fermented with different non-*Saccharomyces* yeast. PCA and variable importance for predictive components (VIP) plots were analyzed based on PLS modeling. The upward and downward triangles represent the volatile flavor compounds identified by GCMS and GC/IMS. Sc, *S. cerevisiae* monoculture. Sc+Zb, *S. cerevisiae* and *Z. bailii* co-inoculation. Sc+Rm, *S. cerevisiae* and *R. mucilaginosa* co-inoculation. Sc+Km, *S. cerevisiae* and *K. marxianus* co-inoculation. Sc+Dh, *S. cerevisiae* and *D. hansenii* co-inoculation. FAJ, apple juice.

**Figure 5 foods-12-03565-f005:**
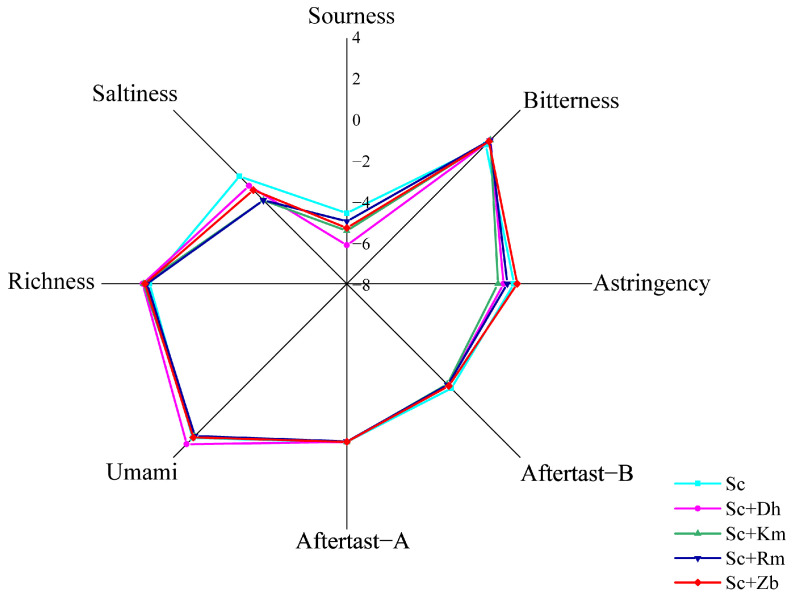
E-tongue profiles of cider samples fermented with different non-*Saccharomyces* yeasts. Sc, *S. cerevisiae* monoculture. Sc+Zb, *S. cerevisiae* and *Z. bailii* co-inoculation. Sc+Rm, *S. cerevisiae* and *R. mucilaginosa* co-inoculation. Sc+Km, *S. cerevisiae* and *K. marxianus* co-inoculation. Sc+Dh, *S. cerevisiae* and *D. hansenii* co-inoculation.

**Table 1 foods-12-03565-t001:** Physicochemical parameters of apple juice and cider samples fermented with different non-*Saccharomyces* yeast.

Samples ^1^	FAJ	Ciders				
		Sc	Sc+Zb	Sc+Rm	Sc+Km	Sc+Dh
Glucose (g/L)	140.00 ± 0.00 ^c^	0.50 ± 0.08 ^a^	0.45 ± 0.07 ^a^	0.30 ± 0.00 ^b^	0.29 ± 0.02 ^b^	0.33 ± 0.03 ^b^
Alcohol (%, *v*/*v*)	ND	10.03 ± 0.76 ^ab^	9.17 ± 0.06 ^b^	10.47 ± 0.68 ^a^	9.37 ± 0.29 ^b^	9.43 ± 0.40 ^b^
pH	3.76 ± 0.02 ^d^	3.94 ± 0.01 ^c^	4.06 ± 0.02 ^b^	3.98 ± 0.03 ^a^	4.03 ± 0.04 ^b^	4.03 ± 0.05 ^b^
TA (g/L)	5.37 ± 0.07 ^b^	6.40 ± 0.33 ^c^	5.16 ± 0.44 ^b^	5.94 ± 0.15 ^a^	5.28 ± 0.09 ^b^	4.89 ± 0.22 ^b^
Glycerol (g/L)	5.97 ± 0.36 ^a^	6.25 ± 0.38 ^a^	6.24 ± 0.66 ^a^	6.48 ± 0.36 ^a^	6.25 ± 0.47 ^a^	6.34 ± 0.40 ^a^
Total yeast count (log CFU/mL)	ND	6.79 ± 0.01 ^a^	6.78 ± 0.01 ^a^	6.79 ± 0.01 ^a^	6.19 ± 0.04 ^c^	6.62 ± 0.02 ^b^

^1^ Results represent the mean ± SD for three independent experiments. Different letters represent significant differences at the 95% confidence level (Duncan’s test). ND = not detected. TA = total acid (expressed as a percentage of tartaric acid). Sc, *S. cerevisiae* monoculture. Sc+Zb, *S. cerevisiae* and *Z. bailii* co-inoculation. Sc+Rm, *S. cerevisiae* and *R. mucilaginosa* co-inoculation. Sc+Km, *S. cerevisiae* and *K. marxianus* co-inoculation. Sc+Dh, *S. cerevisiae* and *D. hansenii* co-inoculation. FAJ, apple juice.

## Data Availability

The data presented in this study are available on request from the corresponding author.

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
