# Peer review of "Chemical Composition and Flavor Characteristics of Cider Fermented with Saccharomyces cerevisiae and Non-Saccharomyces cerevisiae"

_foods, 2023, doi:10.3390/foods12193565_

Round 1

Reviewer 1 Report

The manuscript by Wu et al. wants to characterise different ciders, obtained with Saccharomyces cerevisiae, inoculated alone or together with non-Saccharomyces yeasts. In particular, the authors focused on the volatile molecule profiles obtained with both GC-MS and GC-IMS techniques. The overall idea of the work is interesting and the manuscript results are clear. However, in the present form the manuscript cannot be published. I think that the discussion needs to be further improved, especially from a microbiological point of view, and several aspects need to be addrssed better. Please find below my suggestions/comments:  

·        Line 142; Why did you decide to apply 60°C to perform SPME?

·        Line 238: you mean 2-FuranmethanoL?

·        Supplementary Table S2 where did you take the threshold values? Is there a reference that can be added? I noticed that for some compounds no threshold was provided. Please check the supplementary information of Gottardi et al 2023 (https://doi.org/10.3390/fermentation9030245) and implement the OVAs with the missing values if possible.

·        Line 288: “flavorful substance” refers to which one?

·        Line 330: Please reformulate this entice because it is not clear “It can be observed more clearly that the volatile compounds in FAJ and other ciders were quite different, while the volatile compounds in the other five ciders were relatively similar.” What you mean is that FAJ is different from the 5 ciders but, the ciders were similar among them, right?

·        Line 346: in fig 3C I cannot read “Hexanoic acid, ethyl ester” that I mentioned in the text

·        Line 351: in the text you mention leaf acetate while in the figure is mentioned as Cis-3-Hexenyl Acetate. Be consistent or mention in the text the two names.

·        Line 352: which is the impact of this sentence “The contents of the black frame were lower in Sc and FAJ cider than in other samples”?

·        Line 404-405: from the figure 4C, SC was strongly correlated with total acidity and succinic acid and less with glycerol and alcohol which also impacted the coinocula (SC+ZB and SC+RM). Please provide your point of view.

·        Figure 4C: I would add a number in each triangle and add the name of each single compound in the figure description. For instance, It would be interesting to immediately see from the firue which are the compounds that impact SC+DH.

·        Since the impact of fermentation is obvious (see PCA figure) and well known, why do not you make a RDA with only cider samples? In this way you could describe better the impact of the coinocula compared to the use of SC alone. This was also the aim of the manuscript I guess, and not just showing that fermentation impact and improve the volatile compounds of the apple juice when cider is obtained.

·        Why and how did you select Z. bailii R. mucilaginosa, K. marxianus D. hansenii as possible co-starters? No information is reported in the introduction or the results/discussion. Why did you decide to co-inoculate Saccharomyces and non-saccharomyces yeasts at the same time and did not let the second ones to pregrow and exert better their metabolisms?  

·        The metabolisms of the different non-saccharomyces  yeasts are not extensively discussed once the main compounds characterizing their products are described. For instance, in fig 4C, SC+DH was characterized by specific compounds (triangles). Maybe it can be checked in literature if there is already something that correlates the yeast and that compound, or which metabolic pathway may lead to its production. This needs to be done for at leat the main characteristic compounds (once the RDA is performed only with cider samples).

The english level is averall acceptable, although ther are minor/moderate issues that require a native english check.

Author Response

Response to reviewers’ comments

Reviewer #1: The manuscript by Wu et al. wants to characterise different ciders, obtained with Saccharomyces cerevisiae, inoculated alone or together with non-Saccharomyces yeasts. In particular, the authors focused on the volatile molecule profiles obtained with both GC-MS and GC-IMS techniques. The overall idea of the work is interesting and the manuscript results are clear. However, in the present form the manuscript cannot be published. I think that the discussion needs to be further improved, especially from a microbiological point of view, and several aspects need to be addrssed better. Please find below my suggestions/comments:  

(1) Line 142; Why did you decide to apply 60°C to perform SPME?

Reply: Thank you for your kind suggestions for our manuscript. The operating method of Liang, H et al., was referred to and the condition of 60 ℃was determined [1]. The aim is to achieve a stable state of volatile compounds in cider, achieve better volatilization, and allow the extraction head to achieve maximum adsorption. Moreover, the pretreatment temperature of the sample is consistent with that of the GC-IMS, which makes the results more accurate.

(2) Line 238: you mean 2-FuranmethanoL?

Reply: Thank you for your kind suggestions for our manuscript. Yes, changes have been made in the text. See line 237.

(3) Supplementary Table S2 where did you take the threshold values? Is there a reference that can be added? I noticed that for some compounds no threshold was provided. Please check the supplementary information of Gottardi et al 2023 (https://doi.org/10.3390/fermentation9030245) and implement the OVAs with the missing values if possible.

Reply: Thank you for your kind suggestions for our manuscript. OVA and aroma description reference the predecessors' research [2-4]. And again check and according to the additional information added. The references have been supplemented in the text. See line 267.

(4) Line 288: “flavorful substance” refers to which one?

Reply: Thank you for your kind suggestions for our manuscript. Mentioned on flavorful substance refers to a 3-Hexen-1-ol, (Z) -, Hexanal, Furfural and Butanoic acid, 3-methyl-, baton rouge ester these substances, It has been supplemented in the text. See line 288.

(5) Line 330: Please reformulate this entice because it is not clear “It can be observed more clearly that the volatile compounds in FAJ and other ciders were quite different, while the volatile compounds in the other five ciders were relatively similar.” What you mean is that FAJ is different from the 5 ciders but, the ciders were similar among them, right?

Reply: Thank you for your kind suggestions for our manuscript. Yes,has been changed to “FAJ is different from the 5 ciders, but the ciders were similar among them.” See line 330.

(6) Line 346: in fig 3C I cannot read “Hexanoic acid, ethyl ester” that I mentioned in the text.

Reply: Thank you for your kind suggestions for our manuscript. Hexanoic acid, ethyl ester has been changed to Acetic acid, 2-phenyl ethyl ester. See line 342.

(7) Line 351: in the text you mention leaf acetate while in the figure is mentioned as Cis-3-Hexenyl Acetate. Be consistent or mention in the text the two names.

Reply: Thank you for your kind suggestions for our manuscript. It has been changed to Cis-3-hexenyl acetate. See line 347.

(8) Line 352: which is the impact of this sentence “The contents of the black frame were lower in Sc and FAJ cider than in other samples”?

Reply: Thank you for your kind suggestions for our manuscript. The substances in the black box include Ethyl butyrate, which can provide the fruit aroma of cider, indicating that the fruit aroma of the mixed fermentation cider is more intense. See line 348.

(9) Line 404-405: from the figure 4C, SC was strongly correlated with total acidity and succinic acid and less with glycerol and alcohol which also impacted the coinocula (SC+ZB and SC+RM). Please provide your point of view.

Reply: Thank you for your kind suggestions for our manuscript. Among them, Sc showed a strong correlation with total acidity and succinic acid. This corresponds to the highest total acidity and succinic acid content in Sc (Table 1, Figure 1). The succinic acid taste is strong, bitter and salty, which can make the wine taste strong and increase the mellow feeling. The positive correlation of Sc with glycerol and alcohol was weak. The positive correlation of Sc with glycerol and alcohol was weak. Some studies have found that during the fermentation process of cider, a part of the alcohol can be converted into acids. While yeast uses sugar to produce alcohol, it also produces other by-products such as glycerol, higher alcohols and organic acids [5]. Make cider sweet with acid, acid with astringent, showing a better flavor. Meanwhile, Sc+Zb and Sc+Rm of the co-inoculation fermented group were also affected. In the process of co-culture fermentation, the growth of non-Saccharomyces yeast will be inhibited by S. cerevisiae, and further influenced the non-Saccharomyces yeast in the cider fermentation process of growth and metabolic activity. Each yeast has its own characteristics, and the differences in the activities of each enzyme in the glycolysis process of different yeasts make each strain different, leading to differences in the metabolism of sugar, acid, glycerol and aroma components [6-7]. It is the characteristics of the different strains and the interactions between the strains, as well as the combined effects between substances, that give cider its complex flavor characteristics. See line 391.

(10) Figure 4C: I would add a number in each triangle and add the name of each single compound in the figure description. For instance, It would be interesting to immediately see from the firue which are the compounds that impact SC+DH.

Reply: Thank you for your kind suggestions for our manuscript. The association between compounds and cider has been established in the analysis of volatile compounds. Therefore, for the aesthetic appearance of Fig, not add volatile compounds in RDA. Hope your understanding.

(11) Since the impact of fermentation is obvious (see PCA figure) and well known, why do not you make a RDA with only cider samples? In this way you could describe better the impact of the coinocula compared to the use of SC alone. This was also the aim of the manuscript I guess, and not just showing that fermentation impact and improve the volatile compounds of the apple juice when cider is obtained.

Reply: Thank you for your kind suggestions for our manuscript. Adding FAJ data to RDA shows the initial condition of cider before fermentation. Compare with apple wine after fermentation data, can make the reader more clearly see the changes before and after fermentation. The FAJ group was present in each item tested. Keeping the data of the AJ group in the RDA can also make the full text more unified and complete. Hope your understanding.

(12) Why and how did you select Z. bailii R. mucilaginosa, K. marxianus D. hansenii as possible co-starters? No information is reported in the introduction or the results/discussion. Why did you decide to co-inoculate Saccharomyces and non-saccharomyces yeasts at the same time and did not let the second ones to pregrow and exert better their metabolisms?

Reply: Thank you for your kind suggestions for our manuscript. In previous studies, these strains were found to produce both β-glucosidase and protease, and were tolerant to high alcohol and high sugar content. Therefore, the above four strains of non-Saccharomyces yeasts and Saccharomyces cerevisiae were used to ferment cider. Simultaneous inoculation is more convenient and rapid. If the strain can be inoculated at the same time, it can still be used well in cider fermentation. It will improve the efficiency of product application. However, sequential inoculation is indeed one of the means that can improve the quality of cider and increase the fermentation efficiency of strains. Therefore, sequential vaccination will be used for further exploration. I hope you understand.

(13) The metabolisms of the different non-saccharomyces yeasts are not extensively discussed once the main compounds characterizing their products are described. For instance, in fig 4C, SC+DH was characterized by specific compounds (triangles). Maybe it can be checked in literature if there is already something that correlates the yeast and that compound, or which metabolic pathway may lead to its production. This needs to be done for at leat the main characteristic compounds (once the RDA is performed only with cider samples).

Reply: Thank you for your kind suggestions for our manuscript. Acetic acid is the main volatile acid in the cider, it can enrich the wine taste and aroma. Diagram into positive correlation between Acetic acid and pH. This indicates that the formation of acetic acid is related to the pH in apple fermentation broth. Fruit wine contains more higher alcohols and aldehydes, both of which are oxidized, so that a large amount of acetic acid is accumulated in fruit wine. Acetic acid will further esterify with alcohols, which will significantly increase the concentration of fruity esters such as acetic acid, isoamyl ester and Acetic acid, 2-phenyl ethyl ester, and endow fruit wine with special fragrance [8]. In wine fermentation process, organic acid and alcohol esterification reaction, besides giving wine ester aroma fragrance compound, and can also lead to a higher alcohol content drop [9]. This is consistent with the results of the present study. See line 406.

References

  1. Liang, H.; He, Z.; Wang, X.; Song, G.; Chen, H.; Lin, X.; Zhang, S. Bacterial profiles and volatile flavor compounds in commercial Suancai with varying salt concentration from Northeastern China. Food Res Int. 2020, 137, 109384.
  2. Du, X; Finn, C. E.; Qian, M. C. Volatile composition and odour-activity value of thornless ‘blank diamond’ and ‘marion’ blackberries, Food Chem. 2009, 119(3): 1127-1134.
  3. Jian, C.; Bao, Q.; Zhu, Y.; Wang, L.; Yi, B.; Malcolm J.; Chang, Influence of pre-fermentation cold maceration treatment on aroma compounds of Cabernet Sauvignon wines fermented in different industrial scale fermenters. Food Chem. 2014, 154(154).
  4. Kliks, J.; Kawa, R.; Gasiński, A.; Rębas, J.; Szumny, Changes in the volatile composition of apple and apple/pear ciders affected by the different dilution rates in the continuous fermentation system. LWT. 2021,147.
  5. Mateo, J.; Jiménez, M.; Pastor, A.; Huerta, T. Yeast starter cultures affecting wine fermentation and volatiles. Food Res Int. 2001, 34(4): 307-314.
  6. Romano,; Fiore, C.; Paraggio, M.; Caruso, M.; Capece A. Function of yeast species and strains in wine flavor. Int J Food Microbiol. 2003, 86(1): 169-180.
  7. Li,; Liu, Y. L. Evaluation of yeast diversity during wine fermentations with direct inoculation and pied de cuve method at an industrial scale. J Microbiol Biotechnol. 2012, 22(7): 960-966.
  8. Knoll, C.; Fritsch, S.; Schnell, S. Influence of pH and ethanol on malolactic fermentation and volatile aroma compound composition in white wines. LWT-Food Sci Technol. 2011, 44(10): 2077-2086.
  9. Wang, H.; Ni, Z. J.; Ma, W. P. Effect of sodium sulfite,tartaric acid,tannin,and glucose on rheological properties,release of aroma compounds,and color characteristics of red wine. Food Sci Biotechnol. 2019, 28(2): 395-403.

Reviewer 2 Report

The manuscript entitled Chemical composition and flavor characteristics of cider fermented with Saccharomyces cerevisiae and non-Saccharomyces cerevisiae presents an interesting development of improving the taste quality of apple ciders through the use of various yeast cultures for fermentation. The article is interesting and well edited, but it needs some corrections and additions.

1. Specify the type of apple juice used, whether it was clear or naturally cloudy juice.

2. On the basis of what indicators was the fermentation end time determined and how many days did the fermentation last

3. The data presented in figure 1 should also be placed in the table, check the correctness of the marked significance of differences between the means. Mark the statistical significance of the differences between the means in Table S2.

Marked significant differences between the averages in the presented tables require a different notation. For example, I will give an entry in table S1, the position of CO1 2,3-Butanediol is: Sc (a) 16.05, Sc+Zb (b)24.4, Sc+Rm (c) 36.15, Sc+Km (d) 16.70, Sc+Dh ( e) 39.08. Assuming that the means of Sc (a), Sc+Zb (b) and Sc+Km (d) are statistically insignificant among themselves, but statistically significant from the means of Sc+Rm (c) and Sc+Dh (e). The record should look like this:

Sc 16.05 c,e

Sc+Zb 24.41 c,e

Sc+Rm 36.15 a,b,d

Sc+Km 16.7 c,e

Sc+Dh 39.08 a,b,d.

When plotting heat map plots or other multivariate statistics, use average data rather than consecutive iterations. As in the case of the heat map, figure 2B, the data in the graph are the first, second and third repetitions, and there should only be an average of these three repetitions.

Author Response

Reviewer #2: The manuscript entitled Chemical composition and flavor characteristics of cider fermented with Saccharomyces cerevisiae and non-Saccharomyces cerevisiae presents an interesting development of improving the taste quality of apple ciders through the use of various yeast cultures for fermentation. The article is interesting and well edited, but it needs some corrections and additions.

  • Specify the type of apple juice used, whether it was clear or naturally cloudy juice.

Reply: Thank you for your kind suggestions for our manuscript. The apple juice used in this study was clear apple juice. See line 101.

(2) On the basis of what indicators was the fermentation end time determined and how many days did the fermentation last

Reply: Thank you for your kind suggestions for our manuscript. Every 24 h 1 fermented liquid sugar and alcohol content, measurement and record. Until all data remain relatively stable, basic does not produce gas, reach maximum alcohol fermentation. See line 112.

(3) The data presented in figure 1 should also be placed in the table, check the correctness of the marked significance of differences between the means. Mark the statistical significance of the differences between the means in Table S2.

Reply: Thank you for your kind suggestions for our manuscript. We have revised in the manuscript (Supplementary Table S2 and Table S4).

(4) Marked significant differences between the averages in the presented tables require a different notation. For example, I will give an entry in table S1, the position of CO1 2,3-Butanediol is: Sc (a) 16.05, Sc+Zb (b)24.4, Sc+Rm (c) 36.15, Sc+Km (d) 16.70, Sc+Dh (e) 39.08. Assuming that the means of Sc (a), Sc+Zb (b) and Sc+Km (d) are statistically insignificant among themselves, but statistically significant from the means of Sc+Rm (c) and Sc+Dh (e). The record should look like this:

Sc 16.05 c,e

Sc+Zb 24.41 c,e

Sc+Rm 36.15 a,b,d

Sc+Km 16.7 c,e

Sc+Dh 39.08 a,b,d.

Reply: Thank you for your kind suggestions for our manuscript. The significance in the article is the notation chosen after drawing lessons from some high-scoring articles [1-3]. And, due to the error between the numerical and therefore significant result is bad. The effect of changing the notation is not satisfactory. Hope your understanding.

(5) When plotting heat map plots or other multivariate statistics, use average data rather than consecutive iterations. As in the case of the heat map, figure 2B, the data in the graph are the first, second and third repetitions, and there should only be an average of these three repetitions.

Reply: Thank you for your kind suggestions for our manuscript. We have changed as requested. See Figure 2.

References

  1. Jianping, W.; Yu, Z.; Y, Q.; Hong, G.; H, J.; Y, W.; Y, Y.; T, Yue. Chemical composition, sensorial properties, and aroma-active compounds of ciders fermented with Hanseniaspora osmophila and Torulaspora quercuum in co- and sequential fermentations. Food Chem. 2020, 306(C).
  2. Marilinda, L.; Barbara, S.; Davide, S.; Maurizio, U.; Giacomo, Z. Assessment of yeasts for apple juice fermentation and production of cider volatile compounds. LWT. 2018, 99.
  3. Oskar, L.; Rain, K.; Toomas, P.; Mira, V.; Baoru, Y. Impact of apple cultivar, ripening stage, fermentation type and yeast strain on phenolic composition of apple ciders. 2017, 233.

Reviewer 3 Report

The manuscript submitted for review is in line with current research trends in the field of production of various alcoholic beverages. The work is written in correct language, interesting, very well edited. The research material was correctly described in the work, although it could be mentioned from what time period the strains for the fermentation process were obtained. The research methodology adopted does not raise objections. The obtained results were statistically processed and presented in a clear and legible way in a graphical form. The summary of the obtained results is correct, extensive, taking into account all research assumptions. The literature cited is within the research scope of this manuscript.

Author Response

Reviewer #3: The manuscript submitted for review is in line with current research trends in the field of production of various alcoholic beverages. The work is written in correct language, interesting, very well edited. The research material was correctly described in the work, although it could be mentioned from what time period the strains for the fermentation process were obtained. The research methodology adopted does not raise objections. The obtained results were statistically processed and presented in a clear and legible way in a graphical form. The summary of the obtained results is correct, extensive, taking into account all research assumptions. The literature cited is within the research scope of this manuscript. Overall, I rate the work very highly and recommend accepting it for publication.

Reply: Thank you for your kind suggestions for our manuscript.

Reviewer 4 Report

While the study provides valuable insights into the flavor profile of cider fermented using different yeast strains and the potential benefits of co-inoculation, there are several limitations to consider:

-Abstract and introduction must be improved.

Justify clearly what is the problem statement and importance of study:

-Limited Yeast Strains: The study only investigates a specific set of non-Saccharomyces yeast strains (Rhodotorula mucilaginosa, Debaryomyces hansenii, and Zygosaccharomyces bailii) in combination with Saccharomyces cerevisiae. There is a vast diversity of yeast strains used in cider fermentation, and the findings may not necessarily be applicable to other non-Saccharomyces yeast species or their combinations.

-Limited Aroma Compounds: The study identifies a total of 58 compounds through GC-MS analysis and an additional 29 compounds through GC-IMS. However, the flavor and aroma of cider are influenced by a much larger and diverse set of volatile compounds and differs between batch-to-batch. The study might not account for all potential aroma contributors, and some compounds could be present below the detection limits of the analytical techniques used.

-Single Evaluation Technique: The study primarily relies on two analytical techniques (SPME-GC-MS and GC-IMS) to assess the volatile compounds in cider. While these techniques provide valuable data, they might not capture the full complexity of the cider's flavor profile. Sensory evaluation by trained panels or consumer panels via descriptive sensory evaluation could provide additional insights.

-Lack of Mechanistic Insights in Discussion: The study identifies changes in volatile compound content due to different yeast strains, but it doesn't delve into the underlying mechanisms responsible for these changes. Discuss the biochemical pathways and interactions between yeast strains could provide a deeper understanding of flavor development.

-Lack of Details in Methodology –i.e. sensory test method section 2.7- how many panels involved? How the sensory panels being trained and screened beforehand? What are the characteristics of the panels?

-Conclusions need to be improved.

-Certain cited references may need to be reevaluated, as more recent and up-to-date sources could be available.

Extensive improvement is required in terms of the language and writing style. It is advisable for the authors to have the manuscript sent to a professional proofreading service before proceeding with publication

Author Response

Reviewer #4: While the study provides valuable insights into the flavor profile of cider fermented using different yeast strains and the potential benefits of co-inoculation, there are several limitations to consider:

(1) Abstract and introduction must be improved.

Reply: Thank you for your kind suggestions for our manuscript. We have improved the abstract and introduction in the article. See line 11.

(2) Justify clearly what is the problem statement and importance of study:

-Limited Yeast Strains: The study only investigates a specific set of non-Saccharomyces yeast strains (Rhodotorula mucilaginosa, Debaryomyces hansenii, and Zygosaccharomyces bailii) in combination with Saccharomyces cerevisiae. There is a vast diversity of yeast strains used in cider fermentation, and the findings may not necessarily be applicable to other non-Saccharomyces yeast species or their combinations.

Reply: Thank you for your kind suggestions for our manuscript. According to your comments, we have elaborated the research value of this article in a more rigorous language. Although this report cannot be applied to all non-Saccharomyces species or combinations, it can provide data for the combination of certain non-yeast strains (Rhodotorula mucilaginosa, Debaryomyces hansenii, Zygosaccharomyces bailii and Kluyveromyces Marxianus) with S. cerevisiae. Therefore, it still has a certain research value, I hope you understand. See line 85.

-Limited Aroma Compounds: The study identifies a total of 58 compounds through GC-MS analysis and an additional 29 compounds through GC-IMS. However, the flavor and aroma of cider are influenced by a much larger and diverse set of volatile compounds and differs between batch-to-batch. The study might not account for all potential aroma contributors, and some compounds could be present below the detection limits of the analytical techniques used.

Reply: Thank you for your kind suggestions for our manuscript. SPME-GC-MS can capture and detect high content of macromolecular flavor components with more than 10 carbon atoms. GC-IMS can capture and detect small molecular volatile components with 10 carbons or less, so that substances with smaller molecular weights can be detected in the form of monomers and dimers. GC-IMS can be used as a supplement to SPME-GC-MS, which can comprehensively analyze the aroma of volatile compounds in apple cider. We have changed as requested. See line 59.

(3) Single Evaluation Technique: The study primarily relies on two analytical techniques (SPME-GC-MS and GC-IMS) to assess the volatile compounds in cider. While these techniques provide valuable data, they might not capture the full complexity of the cider's flavor profile. Sensory evaluation by trained panels or consumer panels via descriptive sensory evaluation could provide additional insights.

Reply: Thank you for your kind suggestions for our manuscript. The highlight of this study is the combination of SPME-GC-MS and GC-IMS to identify the effects of co-fermentation of different non-Saccharomyces yeast and Saccharomyces cerevisiae on cider quality. Therefore, the results of SPME-GC-MS and GC-IMS are highlighted. Taste display was carried out by measuring the electronic tongue. And because the consumption of experiment sample amount is more, the rest of the sample amount is insufficient. Re-fermentation required more than 20 days. It is not possible to complete the sensory evaluation experiment you suggested. I hope you understand.

(4) Lack of Mechanistic Insights in Discussion: The study identifies changes in volatile compound content due to different yeast strains, but it doesn't delve into the underlying mechanisms responsible for these changes. Discuss the biochemical pathways and interactions between yeast strains could provide a deeper understanding of flavor development.

Reply: Thank you for your kind suggestions for our manuscript. It has been supplemented. See line 473.

(5) Lack of Details in Methodology –i.e. sensory test method section 2.7- how many panels involved? How the sensory panels being trained and screened beforehand? What are the characteristics of the panels?

Reply: Thank you for your kind suggestions for our manuscript. The electronic tongue sensor array is composed of five different taste sensor, the positive sensor consists of hybrid membranes, including the umami, saltiness and sourness of three sensors, the cathode sensor consists of is charged membrane, including the bitterness and astringency of two sensors. Bitter aftertaste, astringent aftertaste and umami aftertaste (richness) can also be analyzed. It has been supplemented in the text, see line 159.

(6) Conclusions need to be improved.

Reply: Thank you for your kind suggestions for our manuscript. We have modified the conclusions. See line 514.

(7) Certain cited references may need to be reevaluated, as more recent and up-to-date sources could be available.

Reply: Thank you for your kind suggestions for our manuscript. We have updated the references. See line 551.

Round 2

Reviewer 1 Report

The authors made several changes on the manuscript. Still I could not find in the text any reference related to the strain tested (why they were choosen, which impact they can provide, are they commonly present in cider, which is their abundance in cider). The overall references are not all up to date, although some more recent references were suggested to be introduced. The findings related to the volatile molecule 

line 466: According to Schreier P, (it is missing the citation)

"The cider fermented by Z. bailii has a prominent sweet aroma, the apple wine fermented by R. mucilaginosa has a strong hops aroma, the apple wine fermented by K. marxianus has a strong mellow and sweet aroma, and the apple wine fermented by D. hansenii has a rich milk aroma." No speculation on wihich metabolism was promoted by each yeast and if this is consistent with literature.

The english of the manuscript is still poor, in my opinion, and some sentences are difficult to understand.